

# Detection and quantification of gas-phase oxidized mercury compounds by GC/MS

C. P. Jones[1], S. N. Lyman[1], D.A. Jaffe[2], T. Allen[1], T. L. O'Neil[1]

[1]Bingham Research Center, Utah State University, Vernal, 84078, U.S.A.
[2]University of Washington-Bothell, 98011, U.S.A.

*Correspondence to*: S. N. Lyman (seth.lyman@usu.edu)

**Abstract.** Most mercury pollution is emitted to the atmosphere, and the location and bioavailability of deposited mercury largely depends on poorly understood atmospheric chemical reactions that convert elemental mercury into oxidized mercury compounds. Current measurement methods do not speciate oxidized mercury, leading to uncertainty about which mercury compounds exist in the atmosphere and how oxidized mercury is formed. We have developed a gas chromatography/mass spectrometry-based system for identification and quantification of atmospheric oxidized mercury compounds. The system consists of an ambient air collection device, a thermal desorption module, a cryofocusing system, a gas chromatograph, and an ultra-sensitive mass spectrometer. It was able to separate and identify mercury halides with detection limits low enough for ambient air collection (90 pg), but an improved ambient air collection device is needed. The GC/MS system was unable to quantify HgO or $Hg(NO_3)_2$, and data collected cast doubt upon the existence of HgO in the gas phase.

## 1 Introduction

Mercury (Hg) emitted in the gas phase can remain in the Earth's atmosphere for many months and be transported around the globe (Lindberg et al., 2007). Atmospheric Hg pollution is a global problem, and regulation of Hg emissions exist at the state, national, and international levels (Selin, 2009). Hg pollution arises from a variety of natural and anthropogenic point and nonpoint sources (Gustin et al., 2008; Seigneur et al., 2004). Hg can exist in the atmosphere as elemental Hg ($Hg^0$), or as various oxidized Hg compounds ($Hg^{II}$) (Lyman et al., 2010a). Most mercury is emitted to the atmosphere as $Hg^0$ (Pacyna et al., 2006), but it can be oxidized to $Hg^{II}$ in the atmosphere, and $Hg^{II}$ can be reduced to $Hg^0$ (Hedgecock and Pirrone, 2004). $Hg^{II}$ can be found in both the particulate-bound ($Hg^{II}_p$) and gaseous forms ($Hg^{II}_g$) (Sprovieri et al., 2010) and is water-soluble and semi-volatile (Gustin et al., 2008; Lindberg et al., 2007). As a result, aerosols and clouds readily absorb $Hg^{II}_g$, and it is also readily dry deposited (Holmes, 2012; Lyman et al., 2007). Lyman and Jaffe (2012) and others (Gratz et al., 2015; Slemr et al., 2008; Talbot et al., 2008) report that the upper troposphere and lower stratosphere are depleted in $Hg^0$ and enriched in $Hg^{II}$, and oxidation of $Hg^0$ to $Hg^{II}$ has also been shown to occur in the marine boundary layer (Wang et al., 2014) and the Arctic during springtime (Steffen et al., 2008). The location and timing of Hg deposition to ecosystems depends on atmospheric chemistry and the form of Hg in the air (Gustin et al., 2013b; Holmes et al., 2010; Lin et al., 2006; Lyman and Gustin, 2008; Lyman et al., 2010a).



In 1974, Johnson and Braman (1974) suggested that oxidized Hg might consist of HgO and/or Hg halides, and most current studies still echo this hypothesis (Ariya et al., 2009; Holmes et al., 2010; Hynes et al., 2009; Lin et al., 2006). HgO could be produced by reaction of $Hg^0$ with ozone (Pal and Ariya, 2004b), OH (Pal and Ariya, 2004a) or $NO_3$ (Sommar et al., 1997). Evidence exists for the involvement of $NO_3$ in

formation of $Hg^{II}$ (Peleg et al., 2015). Some have argued that HgO is likely to exist only as $Hg^{II}_p$ (Calvert and Lindberg, 2005; Shepler and Peterson, 2003) and that oxidation of $Hg^0$ by ozone may produce an $HgO_3$ intermediate which could decompose to HgO on particles (Calvert and Lindberg, 2005) or react with water to form $Hg(OH)_2$ (Tossell, 2006).  Seigneur et al. (1994) suggested that $Hg(OH)_2$ could be produced from the reaction of $Hg^0$ with $H_2O_2$.

Reactive halogen species, especially bromine species, have received attention as potential $Hg^0$ oxidants since the discovery of Hg depletion events associated with "bromine explosions" during Arctic spring (Schroeder et al., 1998; Steffen et al., 2008), and $HgBr_2$ and/or HgBrOH have been hypothesized as products (Holmes et al., 2006).  Halogen radicals have also been implicated as potential Hg oxidants in the marine boundary layer (Laurier et al., 2003), the Dead Sea (Obrist et al., 2010), and the free troposphere

and stratosphere (Holmes et al., 2006).

$Hg^{II}$ emitted from combustion facilities is generally thought to be $HgCl_2$ (Galbreath and Zygarlicke, 2000; Wilcox and Blowers, 2004). Some oxidized Hg in the atmosphere may be methylmercury, but less than three percent of oxidized Hg in rain water is methylated (Lindberg et al., 2007), and it is likely that most atmospheric $Hg^{II}$ is inorganic.

$Hg^0$, $Hg^{II}_p$, and $Hg^{II}_g$ (operationally defined) are measured routinely at dozens of locations around the world. Current measurement methods for $Hg^{II}_g$ have been shown to be biased low, however (Gustin et al., 2013a; McClure et al., 2014), and ozone and water vapor have been implicated as interferences (Huang and Gustin, 2015; Lyman et al., 2010b). These measurements have been calibrated only rarely, and development and regular deployment of a field-deployable calibrator has been called for (Gustin et al.,

2013b; Jaffe et al., 2014). Current $Hg^{II}$ instrumentation is also not species specific, so it does not provide information about the individual compounds that make up measured $Hg^{II}$ (Jaffe et al., 2014). Different species will have different deposition rates, solubility, and bioavailability (Eagles-Smith and Ackerman, 2014; Peterson et al., 2015), so determining the chemical nature of $Hg^{II}$ is a critical research priority (Gustin et al., 2013b; Jaffe et al., 2014; Malcolm and Keeler, 2007). Huang et al. (2013b) have developed a

thermal desorption system to provide chemical information about $Hg^{II}_g$, but other, complimentary methods are also needed.

Unlike the atomic fluorescence method commonly used for measurement of atmospheric Hg, mass spectrometric methods can be used to identify the chemical composition of Hg compounds (Deeds et al., 2015).  Further, gas chromatography/mass spectrometry (GC/MS) can allow for separation of individual Hg

compounds and separation of Hg compounds from non-mercury components of ambient air samples (Babko et al., 2001; Olson et al., 2002). ICP-MS is routinely used for measurement of Hg and Hg isotope



ratios, but the method is only useful for elemental analysis (dos Santos et al., 2009). GC analysis is routinely used for analysis of organic Hg in various media, and for analysis of $Hg^{II}$ in water after alkylation of the inorganic Hg compounds (Cavalheiro et al., 2014). However, alkylation destroys the native structure of Hg compounds and thus does not provide information about their original identity.

Babko et al. (2001) showed that GC/MS could be used to separate and identify Hg halides. They injected a solution of $HgCl_2$ in acetone into a GC/MS with and found good recovery and consistent peaks. The detection limits, however, were much higher than would be practical for ambient air analysis. Olson et al. (2002) used GC/MS to identify $Hg^{II}$ generated by an $MnO_2$ sorbent in simulated flue gas. They used an impinger to trap the Hg in acetonitrile, then evaporatively concentrated the solution before injecting into a

GC/MS. They injected $HgCl_2$ in acetonitrile and observed mass spectra that were clearly indicative of $HgCl_2$. They also injected $Hg(NO_3)_2$ and $Hg(NO_3)_2 \bullet H_2O$ and saw a similar peak and mass spectra to what they observed in the gas that passed through the $MnO_2$ sorbent.

We have developed a GC/MS-based system to quantify and chemically identify Hg compounds. We describe this analytical system in detail, provide first results, and discuss remaining challenges.

**2 Materials & methods**

The GC/MS-based $Hg^{II}$ detection system consisted of a sample collector to concentrate Hg compounds from the ambient atmosphere, a sample desorber to introduce collected compounds into the gas phase, a cryogenic preconcentrator (cryotrap) to focus and inject Hg compounds, a gas chromatograph to separate Hg compounds from each other and from possible interferents, and an ultra-sensitive mass spectrometer to

definitively determine the chemical speciation of Hg compounds (Fig. 1). It also incorporated a permeation system, pyrolyzer, and $Hg^0$ detector to introduce a consistent, quantifiable amount of various Hg compounds to the system in the gas phase. All wetted parts of the system were kept at at least 160°C (except the sample desorber, which was sometimes cooler), and all wetted parts except the GC columns and VICI GC valve rotors were composed of deactivated fused silica-coated stainless steel. The VICI GC valve

rotors were composed of Valcon E (a polyaryletherketone/PTFE composite).

**2.1 Sample collector**

Four collection materials were tested for suitability to concentrate volatile Hg compounds in air samples. Collection materials tested were nylon, polydimethylsiloxane (PDMS), quartz wool, and deactivate fused silica-coated stainless steel. Nylon membranes were Cole-Parmer nylon polyamide membranes (47mm

round, 0.2 μm thick, P/N: EW-36229-04). The PDMS sorption tubes were those for a Gerstel Thermal Desorption Unit (TDU), and were filled with conditioned PDMS foam (P/N: 013758-105-00). Quartz wool-filled tubes were made from 22 cm long × 1.3 cm diameter Perfluoroalkoxy (PFA) tubing that was washed with soap and water, soaked for 24 hours in a 10% nitric acid bath, rinsed with 18.2 MΩ cm⁻¹ water, then dried in a particle-free environment. The tubing was then filled with quartz wool that had been baked at

800°C for two hours. Deactivated fused silica-coated stainless steel was a 1 cm length of 0.3 cm tubing.



Selection of the best collection surface was based on presence of identifiable Hg peaks on the GC/MS with the least amount of signal interference. Hg was introduced to the sample collectors either from the permeation oven or by passing outdoor ambient air through the collectors. Ambient air samples were collected from Peavine Peak (latitude 39.590, longitude -119.929) near Reno, Nevada; at the University of

Nevada, Reno campus in Reno, Nevada (latitude 39.537, longitude -119.805); and Grizzly Ridge (latitude 40.738, longitude -109.484) near Vernal, Utah using quartz wool-filled tubes and nylon membranes. Quartz wool-filled tube ambient air samples were collected by pulling air through the tubes at 30 L min$^{-1}$ for three hours. Some of the tubes were at ambient temperature during collection, while others were kept at 0°C. Nylon membranes were collected by pulling air through the membranes at 1 L min$^{-1}$ for two weeks. More

information about nylon membrane methods is available from Huang et al. (2013a). All ambient air samples were collected during summer months.

### 2.2 Sample desorber

A thermal desorption module was used to reintroduce collected compounds into the gas phase. We constructed this module by connecting a lab oven to an adjustable digital temperature controller. Membrane

samples were placed within a sample desorption chamber inside the oven, while sample collection tubes were connected directly to the desorption flow path. The desorption chamber was stainless steel coated with deactivated fused silica. A constant flow of Ultra High Purity (UHP) Helium (He) acted as a carrier gas to pass volatilized compounds to the cryotrap. The flow of UHP He was controlled at a rate of 30 mL min$^{-1}$. Desorption temperatures in the range of 80-160°C were used.

### 20 2.3 Cryogenic preconcentrator

The cryogenic preconcentrator (cryotrap) was used to focus desorbed compounds prior to introduction into the GC. A Scientific Instrument Services Model 961 GC Cryo-Trap was used with liquid nitrogen as the cryogen. The cryotrap works by enclosing a portion of the GC column in a small metal cylinder. A flow of liquid nitrogen is passed through the small cylinder at a rate determined by a digital temperature controller.

Volatile compounds are retained on the cooled column. After collection, the metal cylinder rapidly heats up via a nichrome wire heating coil, volatilizing concentrated compounds and allowing them to pass into the GC/MS. During this step, the trap temperature is able to increase by 14°C per second. Cryo-trap cooling temperatures tested ranged from -50°C to 30°C, and heating temperatures tested ranged from 170°C to 240°C. The cryotrap was housed in a lab oven. Lab oven temperatures tested ranged from 160°C to 220°C.

A VICI 6-port GC valve (Model 4C6WT) was housed within the lab oven and controlled flow of sample to the cryo-trap, and from the cryo-trap to the GC. A heated line with a deactivated fused silica guard column (0.25 mm internal diameter) was used to connect the cryotrap to the GC.



### 2.4 Gas chromatograph with an ultra-sensitive mass spectrometer

A Shimadzu GC-2010 Plus gas chromatograph was used to separate Hg compounds from each other and from possible interferents. Different GC column types and lengths were used to test optimum conditions for Hg compound separation. A 30-meter low polarity Restek Rxi-5Sil MS column (5% diphenyl/95% dimethyl polysiloxane), a 60-meter ultra-low polarity Supelco SPB-Octyl fused silica capillary column (50% n-octyl/50% methyl siloxane), and a 30-meter non-polar Restek Rxi-1ms column (100% dimethyl polysiloxane) were tested. GC oven temperatures tested ranged from 140°C to 220°C. After passing through the GC column, the compounds of interest moved to a Shimadzu QP2010 Ultra mass spectrometer for detection of unique chemical signatures of Hg compounds in samples. The MS was operated in high sensitivity electron impact ionization mode, and included a direct probe inlet. Small quantities of solid-phase Hg compounds were added directly to the MS via to the direct probe inlet to determine representative mass spectra for the compounds.

### 2.5 Permeation system, pyrolyzer and Hg detector

Permeation tubes were made to generate four $Hg^{II}$ compounds ($HgBr_2$, $HgCl_2$, $Hg(NO_3)^2$, and $HgO$; Sigma-Aldrich, purity 99.9% or greater). These compounds were packed in permeation tubes constructed of thin-wall 0.3 cm diameter FEP tubing with solid polytetraflouroethylene (PTFE) plugs in both ends. The permeable length of each tube was approximately 1 mm. Larger permeation tubes (1.3 cm diameter × 15 cm permeable length) were also tested for $Hg(NO_3)^2$ and $HgO$. Permeation tubes were enclosed within 0.5 cm inner diameter deactivated fused silica-coated stainless steel tubing. UHP He flowed at 30 mL min⁻¹ through the stainless steel tubing and over the permeation tubes. The tubes were house in an oven consisting of an insulated metal box heated to 100 ± 0.1° C. A VICI multiport GC valve (Model CSF6) selected among four available permeation tubes or passed permeation flow to vent. The multiport valve was housed within the same lab oven as the cryotrap.

A pyrolyzer was used to verify permeation rates of Hg compounds. The pyrolyzer consisted of a 2.5 cm diameter × 18 cm length quartz tube packed with quartz wool. The quartz tube was wrapped with nichrome wire that was used with a variable voltage controller to control the temperature of the tube. The pyrolyzer was kept at 800°C to convert Hg compounds to $Hg^0$ as they passed from the permeation oven through the quartz tube. $Hg^0$ concentrations were measured downstream of the pyrolyzer using a Tekran 2537 mercury vapor analyzer.

### 2.6 Hg compound transmission tests

We permeated $HgBr_2$ into a 1 cm diameter PFA manifold to test the ability of different materials to transmit Hg compounds (Fig. S1). The manifold was heated to 100°C. Air scrubbed of Hg via an activated carbon cartridge was drawn through the manifold at 10 L min⁻¹. A tee pulled a 1 L min⁻¹ subset of air from this manifold into a Tekran 2537/1130 speciation system with a KCl-coated denuder, which measured $Hg^0$ and $Hg^{II}$. The tee to the denuder was 100 cm downstream of the point where $HgBr_2$ was added to the



manifold air. One of several 15 cm long × 0.3 cm diameter tubes was placed between the manifold and the Tekran speciation system to test the ability of these tubes to transmit $HgBr_2$. These 15 cm tubes were constructed of different materials, including stainless steel, PFA (as a control, since the entire manifold was PFA), PEEK, and deactivated fused silica-coated stainless steel. Fittings used to secure the 15 cm tubes

were of the same materials as the tubes. The 15 cm stainless steel, PFA, PEEK, and deactivated fused silica-coated stainless steel tubes were each tested for 24 h.

## 3 Results & discussion

We initially identified $HgBr_2$ and $HgCl_2$ via the method utilized by Babko et al. (2001), which was to dissolve the compounds in acetone and inject the solution into a splitless inlet at 200°C. We used a non-

polar 100% PDMS column at 160°C for separation. We were able to separate $HgCl_2$ from $HgBr_2$ with this system (Fig. 2). The relative abundance of Hg isotopes in Fig. 2 is similar to isotopic abundances reported by IUPAC (de Laeter et al., 2004), confirming the identification. The detection limit of $HgCl_2$ analyzed by this method, calculated as 3 times the standard deviation of seven injections near the detection limit, was 9 ng, much too high for ambient air detection of mercury compounds. Replicate injections showed a high

degree of variability (relative standard deviation of 30 to 45%). The manual syringe used for injections became permanently contaminated with $HgBr_2$ and $HgCl_2$ after use, so we expect that much of the observed variability was due to retention and interactions on the syringe walls, as well as the walls of the injection port.

To eliminate the variability created by liquid injections of these very reactive compounds, we developed

the system described in Methods. We tested a number of different configurations and materials to determine the method most likely to allow for identification of $Hg^{II}$ in ambient air.

### 3.1 System Materials and Temperatures

As a first step in the development of the system, we tested the ability of stainless steel, deactivated fused silica-coated stainless steel (Siltek brand), PFA, and PEEK tubing to transmit gas-phase $HgBr_2$. The lowest

$Hg^{II}$ recovery (and highest $Hg^0$) was observed with stainless steel tubing, followed by PEEK (Fig. 3). Deactivated fused silica-coated stainless steel performed better than PFA, with more $Hg^{II}$ and less $Hg^0$ observed. The amount of total Hg recovered was the same whether deactivated fused silica-coated stainless steel was heated to 135°C or 110-115°C (p = 0.39; the manifold used for these tests was not capable of achieving more than 135°C), but the percent of recovered Hg that was $Hg^{II}$ increased from 83% to 98%. It

is not clear why the 15 cm deactivated fused silica-coated stainless steel tube at 135°C resulted in such an improved $Hg^{II}:Hg^0$ ratio relative to the 15 cm PFA tubing, since a relatively short length of coated stainless tubing was used in a manifold that was otherwise constructed entirely of PFA, and some decomposition of $Hg^{II}$ to $Hg^0$ would be expected in the remainder of the manifold even if the 15 cm tube did not lead to any decomposition. More study of Hg compound decomposition in the presence of different materials is

warranted.



It appears from Fig. 3 that cooler temperatures and less suitable tubing materials led to conversion of $HgBr_2$ to $Hg^0$, perhaps due to reactions facilitated by or on the materials themselves. Others have reported that higher temperatures allow for increased transmission of Hg halides without significant decomposition to $Hg^0$ (Lyman et al., 2010b). Wilcox and Blowers (2004) determined a theoretical temperature-dependent

rate constant for the decomposition of $HgCl_2$ ($HgCl_2 + M \leftrightarrow HgCl + Cl + M$), and compared those results to rate constants developed from experimental data (Widmer et al., 2000). Their theoretical rate equation predicts that $HgCl_2$ will be stable up to about 850°C, while the rate equation determined from experimental data predicts $HgCl_2$ stability up to 450°C (stable = less than half of initial $HgCl_2$ decomposed within 60 s). At 240°C (the maximum temperature reached by any part of our GC/MS system), both rate equations

predict <<1% decomposition within 60 s. Also, L'Vov (1999) showed minimal decomposition of HgO up to 450°C.

We constructed our GC/MS system using deactivated fused silica-coated stainless steel where possible. After the system was constructed, we tested different system temperature settings, including temperatures of the oven that housed the VICI valves and the cryotrap, the temperature of the transfer line from the

cryotrap to the GC, and the GC oven temperature. Table 1 shows that higher peak area and peak height were observed for higher GC oven, valve oven and transfer line temperatures, up to 200°C. These temperatures were optimized for $HgBr_2$, and additional work is needed to determine optimal system temperatures for other Hg compounds.

### 3.2 Cryogenic preconcentrator

The cryogenic preconcentrator (cryotrap) focused Hg compounds before rapidly desorbing them into the GC/MS. Several cryotrap cooling temperatures were tested (Table 2). While cryotrap cooling temperatures of 0°C and -25°C resulted in similar peak areas, peak heights were greater with 0°C, probably because the higher temperature allowed for more rapid desorption when the cryotrap was heated. Peak area and shape deteriorated at cryotrap cooling temperatures above 0°C. When analyzing ambient air samples,

cryotrapping temperatures slightly above 0°C may be ideal, since they would allow for efficient trapping of Hg compounds, but allow water to pass through. While $HgBr_2$ peaks were smaller when the cryotrap was cooled to -50°C, this lower temperature allowed the cryotrap to efficiently collect $Hg^0$ as well (data not shown).

Hotter cryotrap desorption temperatures resulted in better peak areas for $HgBr_2$ (Table 2), with a desorption

temperature of 240°C resulting in the best peak area and peak shape. Hotter desorption temperatures likely resulted in more rapid volatilization of $HgBr_2$ from the cryotrap, leading to improved peak shape.

### 3.3 Chromatographic columns

We tested three different chromatographic columns for transmission of $HgBr_2$, but we were unable to observe $HgBr_2$ peaks with the Rxi-5Sil MS column or the SPB-Octyl column. We observed consistent

$HgBr_2$ peaks with the Rxi-1ms column. Babko et al. (2001) observed $HgCl_2$ peaks with a low polarity



column similar to the RXi-5Sil MS column we used (DB-5; 5% diphenyl/95% dimethyl polysiloxane). Olson et al. (2002) only observed Hg compound peaks when using a 30-m non-polar phase column (DB1; 100% dimethyl polysiloxane) similar to the Rxi-1ms column we used. Olson et al. (2002) were not able to observe Hg peaks with more polar columns that had polyethylene glycol, cyanopropyl phenyl, or trifluoropropyl phases.

### 3.4 Hg compound detection

The permeation rate for $HgBr_2$, determined by the pyrolyzer and Tekran analyzer system, was 37 pg sec$^{-1}$. After incorporating the optimizations reported above, and after further optimizing the parameters of the mass spectrometer, permeation of 22.2 ng of $HgBr_2$ resulted in a peak area of 1,788,451 and a peak height of 143,238 when the MS was operated in selected ion mode for m/z 362 (Fig. 4). The $HgBr_2$ detection limit for the optimized system in selected ion mode, calculated as three times the standard deviation of replicate low-concentration samples, was 90 pg. The detection limit in scan mode was 300 pg.

A small, poorly formed $Hg^0$ peak was often observed prior to Hg halide peaks in chromatograms (see m/z 202 trace in top of Fig. 4). This $Hg^0$ peak was probably the result of breakdown of Hg halides to $Hg^0$ within the chromatographic column, and its poor shape can be explained by continued breakdown as Hg halides moved through the column. The small size of the peak relative to the Hg halide peak is an indicator that Hg halide decomposition in the column was limited. $Hg^0$ chromatographic peaks during Hg halide injections were not likely the result of $Hg^0$ emitted from permeation tubes, since any $Hg^0$ emitted or formed prior to the cryotrap would have passed through the cryotrap at its typical collection temperature of 0°C.

While $Hg^0$ was dectected from $Hg(NO_3)_2$ and HgO-containing permeation tubes (when the cryotrap cooling temperature was lowered to -50°C), we were unable to observe unequivocal $Hg(NO_3)_2$ or HgO mass spectra when analyzing the output of these permeation tubes with the GC/MS. A consistent peak with a prominent 218 m/z signal was observed at a retention time of about 10 minutes when $Hg(NO_3)_2$ or HgO was permeated (Fig. 5). While m/z 218 is the most abundant expected mass for $HgO^+$, the observed isotope pattern did not indicate Hg. Instead, the mass spectrum for this peak was similar to mass spectra for siloxanes, indicating column or tubing degradation as the source. The absence of Hg in this peak was confirmed by the lack of an $Hg^+$ signal at m/z 202 (Fig. 5).

Lyman et al. (2009) constructed HgO permeation tubes and found that, along with $Hg^0$, an Hg compound was emitted from these tubes that could be collected and analyzed using KCl-coated denuders or cation-exchange membranes. Huang et al. (2013a) showed that, when collected on nylon membranes, the Hg compound emitted from HgO permeation tubes exhibited a thermal desorption profile that was different from those exhibited by $HgCl_2$, $HgBr_2$, and $Hg^0$. They showed that $Hg(NO_3)_2$ permeation tubes also emit a reactive Hg compound. It is not known, however, whether the Hg compound emitted from HgO permeation tubes is HgO. Huang et al. (2013a) proposed that the emitted compound could be $Hg_2O$. Regardless of the chemical identity of the compound, it is possible that we were not able to detect it because it degraded within the system tubing, valves or the chromatographic column. The GC/MS did not





report any Hg signal, including $Hg^0$, when analyzing the output from HgO and $Hg(NO_3)_2$ permation tubes with a cryotrap temperature of 0°C. This indicates either (1) the Hg compound emitted from these permeation tubes degraded prior to or within the cryotrap, allowing $Hg^0$ to pass through the cryotrap and out of the system; (2) the emitted Hg compound is too volatile to collect on a cryotrap at 0°C, or (3) the

emitted Hg compound becomes permanently bound to some part of the analytical system and does not degrade. System temperatures and materials were optimized for $HgBr_2$, and different materials and/or a different temperature regime may improve detection for compounds emitted from HgO and $Hg(NO_3)_2$ permation tubes.

We introduced small quantities of HgO and $Hg(NO_3)_2$ (separately) into the direct injection probe on the MS

to determine a mass spectrum for these compounds. Figure 6 shows these mass spectra. Only $Hg^+$ was observed from the direct injection of HgO, and no significant signal was observed around m/z 218 ($HgO^+$) or m/z 420 ($Hg_2O^+$). This could be because (1) HgO is not volatile enough to produce enough vapor in the ionization chamber of the MS to result in a detectable signal, so only off-gassed $Hg^0$ was observed (as $Hg^+$) or (2) the ionization energy of the MS was so strong that all HgO was broken down to $Hg^+$ within the

ionization chamber. $HgO^+$ was observed, however, when $Hg(NO_3)_2$ was inserted into the direct probe, probably as a breakdown product. This provides evidence that the MS could detect HgO as $HgO^+$ if it did indeed exist in the gas phase. The fact that it was not detected when solid HgO was inserted into the direct probe may indicate that HgO does not have an appreciable gas phase. If this is the case, it is not clear what is emitted from HgO permeation tubes that can be collected on nylon membranes and KCl-coated denuders

(Huang et al., 2013a).

The mass spectrum for $Hg(NO_3)_2$ showed $Hg(NO_3)_2^+$ (m/z 326) and several breakdown products, including $HgNO_3^+$ (m/z 264), and $HgO^+$ (m/z 218). A cluster of peaks around m/z 343 was also observed and could be interpreted as the monohydrate of $Hg(NO_3)_2$. Olson et al. (2002) dissolved $Hg(NO_3)_2$ in acetonitrile and interpreted resultant mass spectra to be caused by reaction of $Hg(NO_3)_2$ with column material, producing

$CH_3HgCl$. Our direct probe results suggest the existence of gas phase $Hg(NO_3)_2$ and do not match the spectra presented by Olson et al. (2002).

**3.5 Laboratory Tests of Sample Collection Materials**

We permeated Hg compounds onto nylon membranes and into PDMS foam-filled tubes, quartz wool-filled tubes, and deactivated fused silica-coated tubes and used the sample desorption oven to transmit collected

Hg compounds onto the cryotrap and then into the GC/MS. When we heated nylon membranes to 100°C or greater in the desorption oven, the prominent observed peak was consistent with dodecanoic acid, a potential degradation product of the nylon material (Carraher, 2014). Dodecanoic acid exhibits prominent peaks at and near m/z 200, and the dodecanioc acid peak intensity was so high as to obscure any Hg peaks when loaded in the laboratory or when sampling ambient air. Huang et al. (2013a) used nylon membranes

to collect Hg compounds and desorbed those compounds into a pyrolyzer and atomic fluorescence





analyzer. The atomic fluorescence instrument only detects $Hg^0$, however, so interference from nylon breakdown products was not an issue.

Like nylon membranes, PDMS foam-filled tubes also had too much interference to allow for detection of Hg compounds, even when loaded with as much as 60 ng $HgBr_2$.

Quartz wool-filled PFA tubes exhibited less interference and clearly identifiable Hg halide signals (Fig. 7). However, chromatograms from quartz wool-filled tubes had poorly-shaped peaks and substantial non-Hg signal. The cause of the poorly-shaped peaks is not known. Quartz particles could have accumulated in the cryotrap, causing a slow desorption of Hg compounds during cryotrap heating. If this occured, the quartz particles were apparently cleaned out after each analysis, since we performed injections of Hg halides

directly from the permeation oven onto the cryotrap subsequent to quartz wool analyses and observed normal chromatographic peaks.

While $HgCl_2$ was loaded onto a quartz wool-filled tube for the analysis shown in Fig. 7, the figure shows some $HgBr_2$ signal in the mass spectrum. The quartz wool-filled tube had previously been loaded with $HgBr_2$, and some residual $HgBr_2$ apparently remained in the tube. Also, a cluster of masses centered at m/z

316, corresponding with HgBrCl, can be clearly observed in the mass spectrum. The same cluster of masses can be seen in part B of Fig. 2 and in Fig. 4. HgBrCl could exist because of (1) a reaction between $HgCl_2$ and $HgBr_2$ in the acetone solution in Fig. 2 or on the quartz wool in Fig. 7, or (2) it could be a contaminant in the commercial Hg halide compounds used in this study.

The deactivated fused silica-coated stainless steel tube collected $HgBr_2$ with no discernable non-Hg

interference, but the amount of $HgBr_2$ collected was low, probably because of high breakthrough. Deactivated fused silica-coated tubing may be a viable $HgBr_2$ collection device if a device with larger surface area is used and/or if the tubing is cooled to a temperature that limits $Hg^{II}$ breakthrough.

**3.6 Ambient Air Sample Collection**

The location, duration, and methods used for ambient air sample collection are given in Methods. None of

the ambient air samples collected using quartz wool-filled tubes or nylon membranes resulted in any detectable Hg compounds. Low $Hg^{II}$ concentrations in ambient air could have led to a sampled $Hg^{II}$ mass below the detection limits of the GC/MS. 10.1 $m^3$ of ambient air were sampled by each nylon membrane, and 5.4 $m^3$ of air were sampled by each quartz wool-filled tube. These sampling volumes are adequate to collect 100 pg $Hg^{II}$ (within the detection limit of the GC/MS) if ambient $Hg^{II}$ was 10 and 18 pg $m^{-3}$,

respectively. Not all ambient air sample collections were associated with alternative $Hg^{II}$ measurements, but for quartz wool-filled tube collection at the University of Nevada, Reno, $Hg^{II}$ (measured by a Tekran 2537/1130/1135 speciation system as the sum of Hg collected on the system's denuder and particulate filter) was $38 \pm 9$ pg $m^{-3}$ (mean ± 95% confidence interval). However, some breakthrough may have occurred through the sample collection devices, and the non-Hg interference caused by the sample

collection devices likely increased the actual detection limit for these samples. Alternatively, it is possible



that the Hg compounds in sampled ambient air were not Hg halides and were undetectable by the GC/MS system.

### 3.7 Future Work

Further improvements to the MS may increase its sensitivity for Hg compounds and decrease the detection

limits of our system. Deeds et al. (2015) reported detection limits of 6-40 pg for $HgCl_2$ and $HgBr_2$ with an atmospheric pressure chemical ionization MS, and pointed out that chemical ionization is likely to decrease detection limits relative to electron impact ionization. The system used by Deeds et al. (2015), however, suffered from interference from non-Hg compounds found in ambient air because it did not utilize chromatographic separation. Our GC system, coupled with chemical ionization MS, may be able to

achieve improved ambient air detection limits while maintaining the ability to separate individual Hg compounds and separate Hg compounds from non-Hg atmospheric constituents.

Interference in mass spectra created by collection materials likely limited our ability to detect Hg compounds in ambient air. Testing of additional collection materials is needed. Deeds et al. (2015) used shredded Teflon packed in tubes to collect Hg halides, and they did not note any interference from these

materials. We found no interference from deactivated fused silica-coated stainless steel tubing. Highly inert surfaces like these are ideal because they do not result in off-gasing that may interfere with mass spectra. However, $Hg^{II}$ may not collect efficiently on these surfaces unless they are cooled to 0°C or lower. Also, Lyman et al. (2010b) reported that ozone reduces Hg halides collected on uncoated quartz traps to $Hg^0$, and highly inert surfaces may also leave $Hg^{II}$ exposed to reaction with ozone or other atmospheric

constituents.

We observed poor results from tubes packed with PDMS foam, but saw little interference from the PDMS-coated chromatographic column (for Hg halides). PDMS in chromatographic columns is cross-linked to stabilize it and is less likely to decompose. PDMS denuders have been used successfully to preconcentrate a wide variety of compounds (Burger et al., 1991; Dudek et al., 2002), including semivolatiles (Rowe and

Perlinger, 2010). PDMS may also shield analytes from atmospheric oxidants that have low affinity for the PDMS phase (possibly including ozone) (Rowe and Perlinger, 2010).

While the GC/MS system in this study was able to separate and quantitatively analyze Hg halides, we have not yet shown that it can detect non-halide Hg compounds, including $Hg(NO_3)_2$ and HgO. HgO may not exist in the gas phase, but $Hg(NO_3)_2$ is likely to be nonpolar, as are Hg halides (Goodsite et al., 2004), and

likely can exist in the gas phase. HgBrX compounds, including HgBrOH, may exist in the atmosphere (Weiss-Penzias et al., 2015). HgBrOH, unlike $HgBr_2$, has an appreciable dipole moment (Goodsite et al., 2004) and may have different reactivity and volatility than $HgBr_2$. Our system is able to detect HgBrCl, so it could possibly identify other bromine-containing Hg compounds, but this has not been tested. System temperatures and materials may need to be optimized for individual compounds or groups of compounds.

Different Hg compounds may perform better with different columns, as shown by Babko et al. (2001). Finally, replacement of VICI valves (which have Valcon rotors that may react with Hg compounds) with



all-stainless steel valves that can be coated with deactivated fused silica may improve system performance for Hg compounds that are more reactive than Hg halides.

## 4 Summary

Identification of atmospheric Hg$^{II}$ is needed to improve understanding of Hg chemistry and biogeochemical
cycling. The GC/MS-based system described here has a detection limit low enough to identify and quantify Hg halides in the ambient atmosphere, but better atmospheric sampling materials are needed to accomplish this. Work is ongoing to continue its development, including improving ambient air collection options, increasing the number of compounds it can reliably detect, and improving instrument sensitivity.

## 5 Acknowledgements

Funding for this work was provided by U.S. National Science Foundation award number 1324781 and the Electric Power Research Institute.  We are grateful to Mae Gustin, Jiaoyan Huang, and Matthieu Miller at the University of Nevada, Reno for collecting some of the ambient air samples used in this study and to Dr. Gustin for providing helpful comments on the manuscript.

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





| GC Oven Temp (°C) | Retention Time | Peak Area | Peak Height |
|---|---|---|---|
| 140 | 5.91 | 11,794 | 802 |
| 160 | 4.52 | 14,676 | 1077 |
| 180 | 3.87 | 17,037 | 1158 |
| 200 | 2.66 | 22,892 | 1235 |
| **Valve Oven and Line Temp (°C)** | | | |
| 180 | 4.32 | 36,058 | 1989 |
| 200 | 4.19 | 40,646 | 1998 |

Table 1. GC/MS results from permeation of $HgBr_2$ with varying GC oven, valve oven, and line temperatures. Each iteration was performed at least in duplicate to verify the consistency of results. Peak area and height are for m/z 362.





| Cryotrap Temp (°C) | Cooling | Retention Time | Peak Area | Peak Height |
|---|---|---|---|---|
| -50 | | 4.21 | 104,386 | 4,789 |
| -25 | | 4.22 | 137,471 | 7,206 |
| 0 | | 4.14 | 134,462 | 8,623 |
| 5 | | 4.14 | 117,864 | 8,272 |
| 10 | | 4.12 | 113,516 | 7,589 |
| 30 | | 4.13 | 39,526 | 1,583 |
| Cryotrap Temp (°C) | Desorb | | | |
| 170 | | 3.35 | 83,198 | 8,582 |
| 200 | | 4.14 | 101,757 | 8,557 |
| 220 | | 3.36 | 173,059 | 17,501 |
| 240 | | 3.377 | 175,723 | 17,567 |

Table 2. GC/MS results from permeation of $HgBr_2$ with varying cryotrap cooling and desorption temperatures. Each iteration was performed at least in duplicate to verify the consistency of results. Peak area and height are for m/z 362.



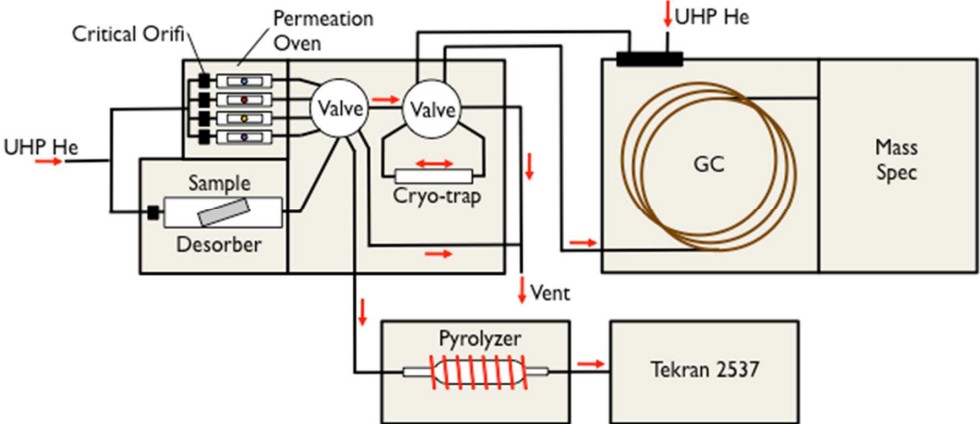

Figure 1. Diagram of GC/MS system used to identify Hg compounds, and a pyrolyzer and Tekran 2537 system to quantify the amount of Hg compounds generated by the permeation oven.



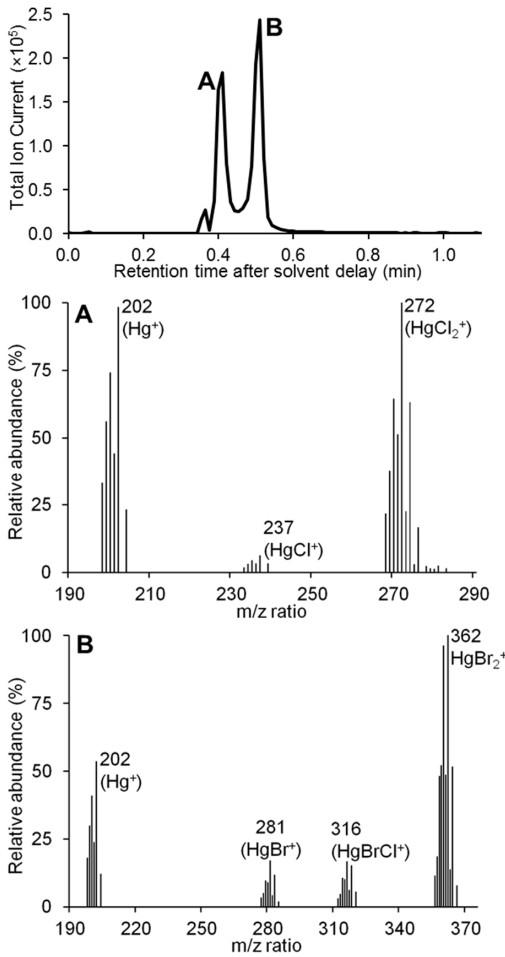

Figure 2. Chromatogram of a mixed standard of $HgCl_2$ and $HgBr_2$ in acetone (0.5 µg µL$^{-1}$ for each compound) (top), mass spectra for $HgCl_2$ (A, middle) and $HgBr_2$ (B, bottom). Solvent delay was 2 min.



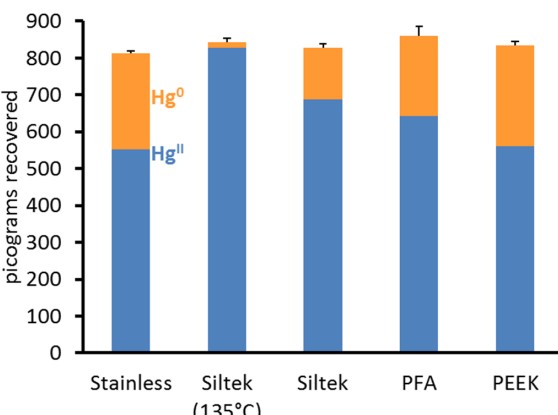

Figure 3. Picograms of $Hg^{II}$ and $Hg^0$ recovered when $HgBr_2$ was permeated into mercury-free air and passed through a 15 cm section of 0.3 cm diameter tubing constructed of the materials indicated. Tubing was kept at 110-115°C, except when indicated otherwise. "Siltek" indicates stainless steel tubing coated with Siltek-brand deactivated fused silica.



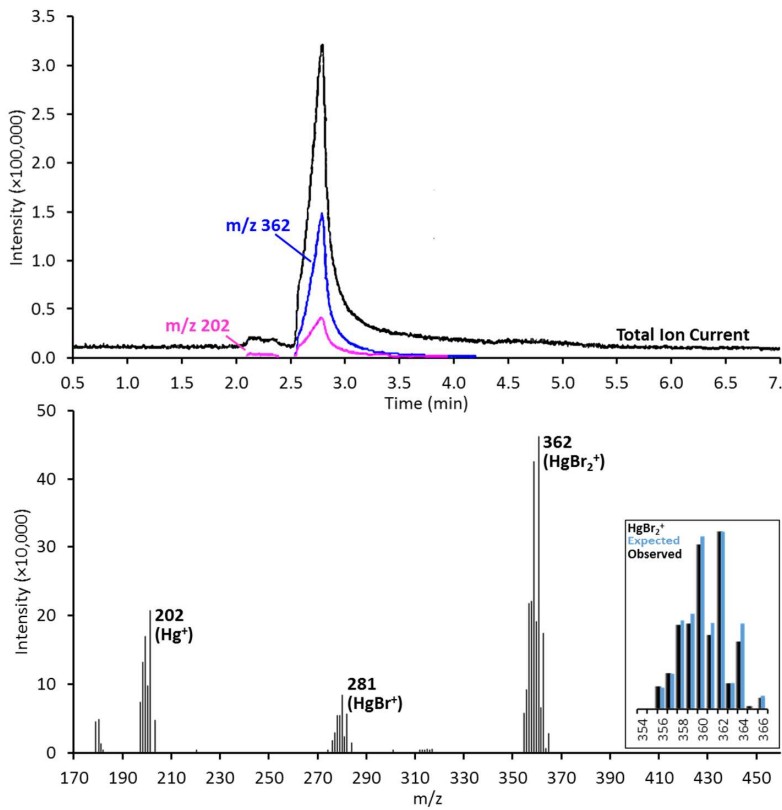

Figure 4. Chromatogram (above) and mass spectrum at 2.92 min (below) of $HgBr_2$ emitted from a permeation tube. The Y-axis shows the mass spectrometer's ion current intensity. The box at bottom left compares the expected $HgBr_2^+$ mass spectrum (from de Laeter et al. (2004)) to the observed spectrum.



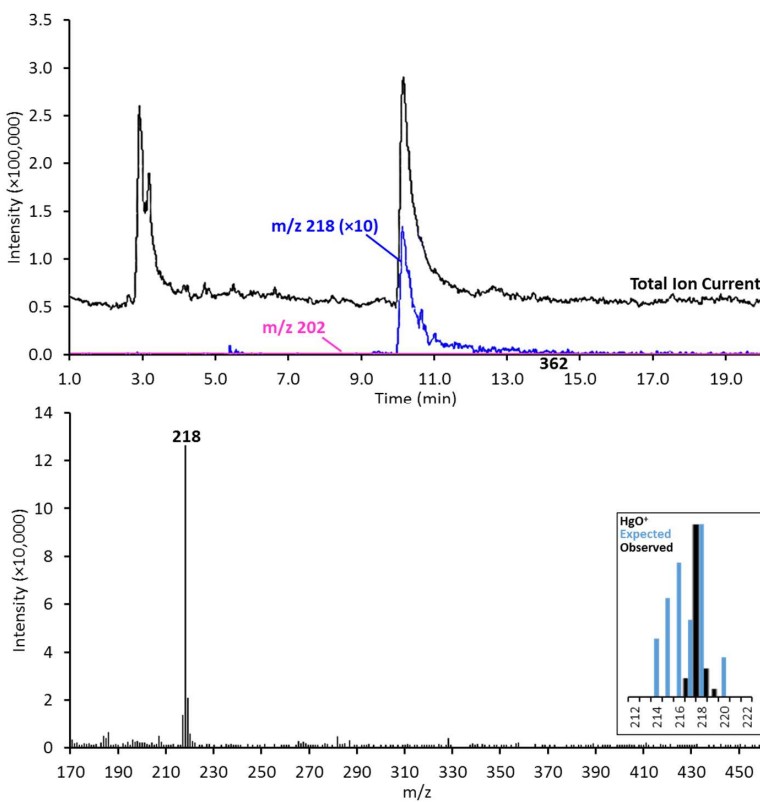

Figure 5. Chromatogram (above) and mass spectrum at 10.10 min (below) generated from analysis of 34 ng Hg emitted from an HgO permeation tube. The box at bottom left compares the expected HgO$^+$ mass spectrum (from de Laeter et al. (2004)) to the observed spectrum. While the prominent mass peak was m/z 218, the observed isotope pattern does not indicate HgO$^+$.





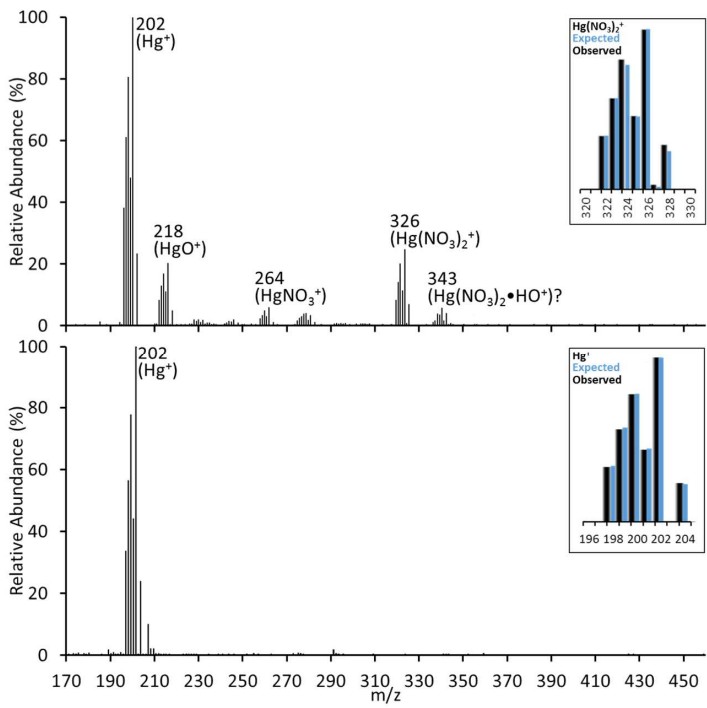

Figure 6. Mass spectrum for $Hg(NO_3)_2$ (above) and HgO (below) derived from direct probe injection into the MS. The boxes at top left in each pane compare the expected indicated mass spectrum (from de Laeter et al. (2004)) to the observed spectrum.




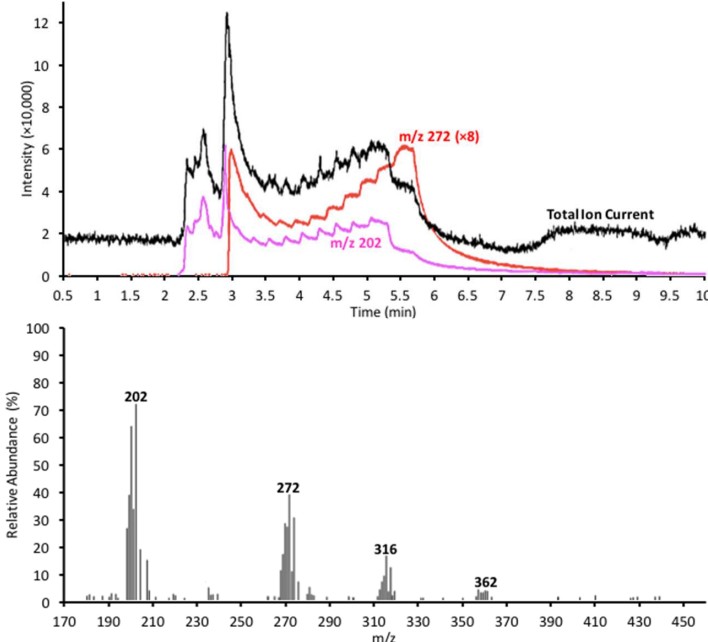

Figure 7. Chromatogram (above) and mass spectrum at 5.6 min (below) of $HgCl_2$ permeated onto quartz wool-filled PFA tubing, then desorbed into the GC/MS system.