# Peer review of "Detection and quantification of gas-phase oxidized mercury compounds by GC/MS"

_Atmospheric Measurement Techniques, 2016_

## Referee Comment (RC1) · Anonymous Referee #2 · 29 Mar 2016

This paper presents the details of a GC-MS method to quantify gaseous HgII compounds which are thought to exist in the atmosphere but as of yet been unquantified. There is currently no method to collect these compounds from ambient air and determine their concentrations. GC-MS appears to be a promising technique and the authors present a nice study of the details involved in generating standards, which materials to use, and what limits were encountered with suggestions for future work.

I found the manuscript clearly written, properly cited, the figures were clear and results clearly explained. I have no major criticisms and found nothing specific for the authors to address.

My only reservation is that the ultimate goal of detecting and quantifying HgII in ambient may not be achievable with this technique. The authors do a good job of not overstating

optimism so I can't fault them for that. Hopefully this initial work will lead to steady improvements in this area.
* * *

---

## Referee Comment (RC2) · Anonymous Referee #1 · 31 Mar 2016

Detection and quantification of gas-phase oxidized mercury compounds by GC/MS

Summary: This paper describes the development of an instrument to hopefully identify and measure atmospheric oxidized mercury compounds. Currently there are no measurement techniques that speciate and quantify oxidized mercury. Much is unknown about the chemical composition of oxidized mercury and knowing the chemical form of mercury is critically important because mercury is a toxic pollutant and the forms that it takes determine its interaction with the environment. This paper gives a nice introduction to the subject for those not familiar with it. It is fairly well organized but the Materials and Methods section(s) and the Results and Discussion section(s) are organized in a similar way and at times it is a little confusing as to what material should be put in the Methods section and what should be put in the Results section. I understand why the authors did this and it works mostly but there are some places for improvement

in the presentation – I try to point out a few of these below.

This is a good contribution to the literature as it points out a possible approach to detecting the oxidized mercury species and also shows how difficult it appears to be.

The analytical methodology is based on good principles and understanding of the scope of the challenge. The wetted parts of the system were kept at least at 160 C which is a challenge in itself. I did not understand the preparation steps for the PDMS sorption tubes. What was trying to be accomplished by the acid was etc.?

In section 3.1 line 27 – 35: this is a very strange result and although the authors acknowledge it as such it really doesn't make much sense. Was the percent HgII recovery always increased from 83 and 98% or was this a one time test? If it represents multiple tests there should be some variability reported in the results.

Page 7 line 12: change to "we constructed the plumbing for our GC/MS system...

Page 7 line 20: Put the first two sentences in the Methods section (2.3) along with Table 2.

Page 7 line 33: Already stated the three columns that were used. Sugget to re-write that first sentence: Of the three columns we tested for transmission of HgBr2, we only observed peaks with....

Page 11 line 6: "Further improvements to the MS..." - what improvements have already been made? – this suggests there were some – and by whom were the improvements made?

Page 11 line 11: It strikes me that perhaps the authors should try this before publishing this paper.

---

## Author Comment (AC1) · 19 Apr 2016

Thank you for this review. We have tried to be careful not to overstate the performance of this instrument. We feel that, since some Hg compounds are fairly easily detectable by GC/MS and others are not, whether detection and identification of Hg compounds in ambient air is possible may depend on the chemical form of those compounds.

---

## Author Comment (AC2) · 19 Apr 2016

We have reproduced the recommendations of the anonymous referee, and have provided our responses after each comment

Anonymous Referee: In section 3.1 line 27 – 35: this is a very strange result and although the authors acknowledge it as such it really doesn't make much sense. Was the percent HgII recovery always increased from 83 and 98% or was this a one time test? If it represents multiple tests there should be some variability reported in the results.

Response: As indicated in section 2.6, each test was performed over 24 hours, so we collected many measurements for each experimental condition. Our Tekran speciation system collected one measurement every 1.5 hours, so we obtained 16 measurements

per experimental condition. We have added wording to this effect to section 2.6. We included 95% confidence intervals in Figure 3, which are a measure of variability in the results. We neglected to include in the figure caption that the whiskers represent 95% confidence intervals, but we have corrected this omission in the revised manuscript.

Anonymous Referee: Page 7 line 12: change to "we constructed the plumbing for our GC/MS system. . .

Response: Reworded the sentence to clarify what was constructed. This is now on Page 7 Line 13.

Anonymous Referee: Page 7 line 20: Put the first two sentences in the Methods section (2.3) along with Table 2.

Response: We omitted the first sentence, and reworded the second sentence. Table 2 shows results of instrument tests and so was left in the Results and Discussion section.

Anonymous referee: Page 7 line 33: Already stated the three columns that were used. Sugget to re-write that first sentence: Of the three columns we tested for transmission of HgBr2, we only observed peaks with. . ..

Response: Reworded this part of Section 3.3

Anonymous Referee: Page 11 line 6: "Further improvements to the MS. . ." - what improvements have already been made? – this suggests there were some – and by whom were the improvements made?

Response: We have reworded this sentence to make it more clear.

Anonymous Referee: Page 11 line 11: It strikes me that perhaps the authors should try this before publishing this paper.

Response: We currently do not have chemical ionization module to add to the MS and perform these tests. This will have to wait until we receive additional funding.

---

## Author Comment (AC3) · 19 Apr 2016

We have attached the revised manuscript with changes tracked.

Please also note the supplement to this comment:
http://www.atmos-meas-tech-discuss.net/amt-2016-29/amt-2016-29-AC3-supplement.pdf